# FIGRDock: Fast Interaction-Guided Regression for Flexible Docking

**Shikun Feng**[1,2]*, **Bicheng Lin**[3,1]*†, **Yuanhuan Mo**[4,1]*†, **Yuyan Ni**[1,5], **Wenyu Zhu**[1], **Bowen Gao**[1], **Wei-Ying Ma**[1], **Haitao Li**[3], **Yanyan Lan**[1,6,7]‡

[1]Institute for AI Industry Research (AIR), Tsinghua University, Beijing, China
[2]Zhongguancun Institute of Artificial Intelligence, China
[3]School of Basic Medical Sciences, Tsinghua University, China
[4]School of Software Engineering, South China University of Technology
[5]Academy of Mathematics and Systems Science, Chinese Academy of Sciences
[6]Beijing Frontier Research Center for Biological Structure, Tsinghua University, Beijing, China
[7] Beijing Academy of Artificial Intelligence, Beijing, China

## Abstract

Flexible docking, which predicts the binding conformations of both proteins and small molecules by modeling their structural flexibility, plays a vital role in structure-based drug design. Although recent generative approaches, particularly diffusion-based models, have shown promising results, they require iterative sampling to generate candidate structures and depend on separate scoring functions for pose selection. This leads to an inefficient pipeline that is difficult to scale in real-world drug discovery workflows. To overcome these challenges, we introduce FIGRDock, a fast and accurate flexible docking framework that understands complicated interactions between molecules and proteins with a regression-based approach. FIGRDock leverages initial docking poses from conventional tools to distill interaction-aware distance patterns, which serve as explicit structural conditions to directly guide the prediction of the final protein-ligand complex via a regression model. This one-shot inference paradigm enables rapid and precise pose prediction without reliance on multi-step sampling or external scoring stages. Experimental results show that FIGRDock achieves up to 100× faster inference than diffusion-based docking methods, while consistently surpassing them in accuracy across standard benchmarks. These results suggest that FIGRDock has the potential to offer a scalable and efficient solution for flexible docking, advancing the pace of structure-based drug discovery.[4]

## 1 Introduction

Molecular docking refers to predicting the three-dimensional structure of a protein–ligand complex given the individual structures of the protein and the small molecule. This task is fundamental to structure-based drug discovery, as it enables large-scale screening and mechanistic understanding of molecular interactions that underlie pharmacological effects. While conventional docking methods typically assume a rigid protein conformation, flexible docking models the conformational adjustments of both the ligand and the protein, especially those arising from induced-fit effects. By capturing this dynamic binding process, flexible docking provides a more biologically realistic framework, though it also introduces significant computational and modeling complexity.

---

*Equal contribution.
†Work was done while Bicheng Lin and Yuanhuan Mo were research interns at AIR.
‡Correspondence to: Yanyan Lan <lanyanyan@air.tsinghua.edu.cn>.
[4]The code is open-sourced in link `https://github.com/fengshikun/FIGRDock.git`

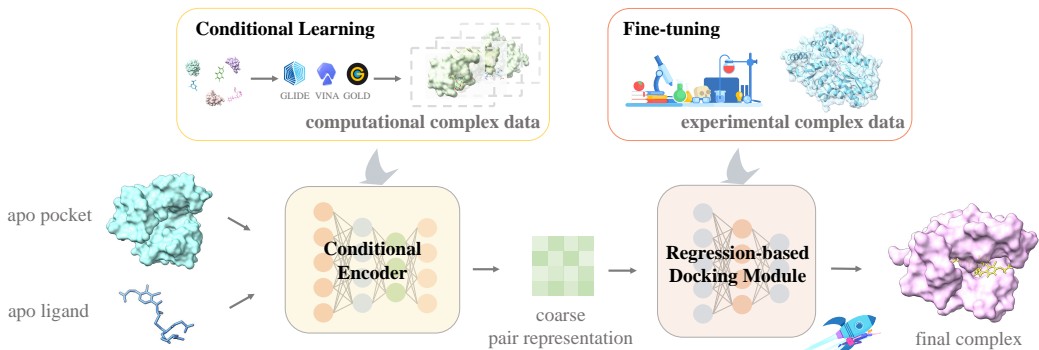

Figure 1: Overview of FIGRDock. The model mainly comprises two modules: a conditional encoder and a regression-based docking module. The conditional encoder, which is pre-trained using computational complex data, aims at providing a coarse pair representation. The regression-based docking module, fine-tuned on more accurate experimental complex data, is tailored to conduct flexible docking with efficiency and accuracy under the guidance of conditional pair representation.

Deep learning has recently brought significant advances to the flexible docking problem, offering data-driven alternatives to traditional physics-based methods. Among these, two major paradigms have emerged: co-folding approaches that predict complex structures directly from protein sequences, and generative approaches that operate on given unbound(apo) protein and ligand structures. Co-folding models, exemplified by AlphaFold 3 [1], achieve impressive accuracy but remain computationally intensive due to the inherent complexity of protein structure prediction. In contrast, generative methods [5, 17, 8, 4] leverage diffusion models to sample ligand poses, including global translation, rotation, and torsion angles of ligand rotatable bonds and protein side chains, conditioned on apo structures. By restricting the generative process to this product space, these methods significantly reduce the dimensionality of the prediction task, and offer substantial improvements in efficiency over co-folding approaches.

Despite these advances, current generative models still face notable limitations that hinder their practical deployment. They typically rely on multi-step sampling to produce accurate protein-ligand complexes and require repeated generation–scoring cycles, where performance improves with more iterations [5, 17, 8]. Moreover, they depend heavily on pre-trained protein language models, such as ESM2 [10], to provide amino acid embeddings that serve as initial node features [16, 5].

In contrast to diffusion-based models, regression-based approaches have been widely adopted in rigid docking frameworks such as EquiBind [19] and TANKBind [12], offering superior efficiency via one-shot pose prediction. However, their performance in flexible docking scenarios is often suboptimal [5], as one-shot inference struggles to capture induced-fit effects and conformational changes in the ligand or protein. This gap raises a compelling question: **Can we design a regression-based docking framework that retains the efficiency of one-shot prediction while achieving high accuracy under flexible docking scenarios?**

To the best of our knowledge, accurate interaction modeling is essential to realize the full potential of regression-based docking. Unlike generative approaches, which refine predictions through multiple iterations, regression models infer binding conformations in a single pass. This one-shot approach demands precise interaction modeling, as there is no iterative correction process like in generative methods. In flexible docking scenarios, where even subtle conformational adjustments are critical, any misrepresentation of interactions can lead to significant deviations in predicted binding poses.

Building on this insight, we propose Fast Interaction-Guided Regression for Docking (FIGRDock). This method directly regresses to an accurate docking complex structure through a single network inference, guided by interaction representations, enabling both higher precision and greater efficiency in the docking process. FIGRDock's training is organized into two stages, as illustrated in Figure 1. The first stage involves conditional pair representation learning. We leverage the SIU dataset [7], which contains a substantial amount of synthetic computational complex data generated by docking software, as pre-training data to learn interaction-informed paired representations between the protein and ligand. Despite the lower precision compared to crystallographic data, computational structures

compensate for the limited availability of experimental structures. In the second stage, this learned pair representation is used as input for the regression-based docking module, followed by fine-tuning on more accurate crystal complex structures. The regression approach requires only a single network inference to predict the structure. Guided by the interaction pair representation, it produces more accurate predicted structures than iterative generative methods.

Experimental results show that when compared to generative methods, FIGRDock reduces inference time from the order of tens of seconds to hundreds of milliseconds—a nearly 100× speedup. Furthermore, by leveraging pair representations as conditions, FIGRDock achieves superior performance across both holo and apo input test scenarios. To the best of our knowledge, this is the first regression-based method to achieve comparable or even better performance than diffusion-based methods. In the context of the dominance of generative models in flexible docking, our work offers a promising alternative approach that could provide valuable insights and solutions for future research in the field.

## 2    Related work

In this section, we briefly review related work on flexible docking, focusing on two main approaches: co-folding methods and diffusion-based generative models.

### 2.1    Co-folding Methods

Co-folding methods aim to predict the three-dimensional structure of protein–ligand complexes in an end-to-end fashion. These approaches take as input a protein sequence and a molecular representation of the ligand, typically in the form of a molecular graph or SMILES string, and directly output the bound complex structure. Recent advances such as NeuralPLexer [18], Umol [2], AlphaFold3 [1], and HelixFold3 [11] have demonstrated the effectiveness of this paradigm, achieving impressive accuracy in modeling protein–ligand interactions from minimal input information. However, the high computational demands of training and inference in these models pose significant challenges, limiting their scalability and practicality for large-scale virtual screening applications.

### 2.2    Diffusion-based Generative Models

Diffusion-based generative models have emerged as a leading paradigm for flexible docking. These methods take as input the unbound (apo) structures of both the protein and ligand and generate the bound complex structure by modeling the joint conformational changes that occur upon binding. Instead of searching over large configuration spaces or simulating the full folding process from the sequence, these models leverage generative diffusion processes to sample binding poses in a data-driven manner. Representative methods such as DiffDock-Pocket [17], DiffBindFR [25], and Re-Dock [8] use diffusion or diffusion-bridge frameworks to capture pocket side-chain flexibility. FlexDock [4] and DynamicBind [13] further incorporate backbone flexibility using techniques such as unbalanced flow matching and geometric diffusion. While these approaches have shown strong accuracy in modeling flexible binding, they often suffer from inefficiencies due to iterative sampling and dependence on external scoring functions for pose selection.

Recently, there have been several initial attempts to alleviate the inefficiency problem. A representative example is FABFlex [23], which directly predicts protein-ligand conformation with a regression model. Unlike diffusion-based methods that rely on iterative sampling, regression models aim to directly predict the final bound structure in a single forward pass, offering significantly improved inference efficiency. FABFlex, which takes the apo ligand and protein backbone as input and regresses the ligand pose along with the $C_\alpha$ coordinates of binding site residues. While this approach greatly reduces computational cost, its accuracy still lags behind state-of-the-art diffusion-based models. Moreover, because it does not explicitly model side-chain flexibility, where much of the binding-induced conformational change occurs, its ability to capture fine-grained interactions remains limited. These limitations motivate the development of more accurate and interaction-aware regression-based approaches, such as our proposed FIGRDock.

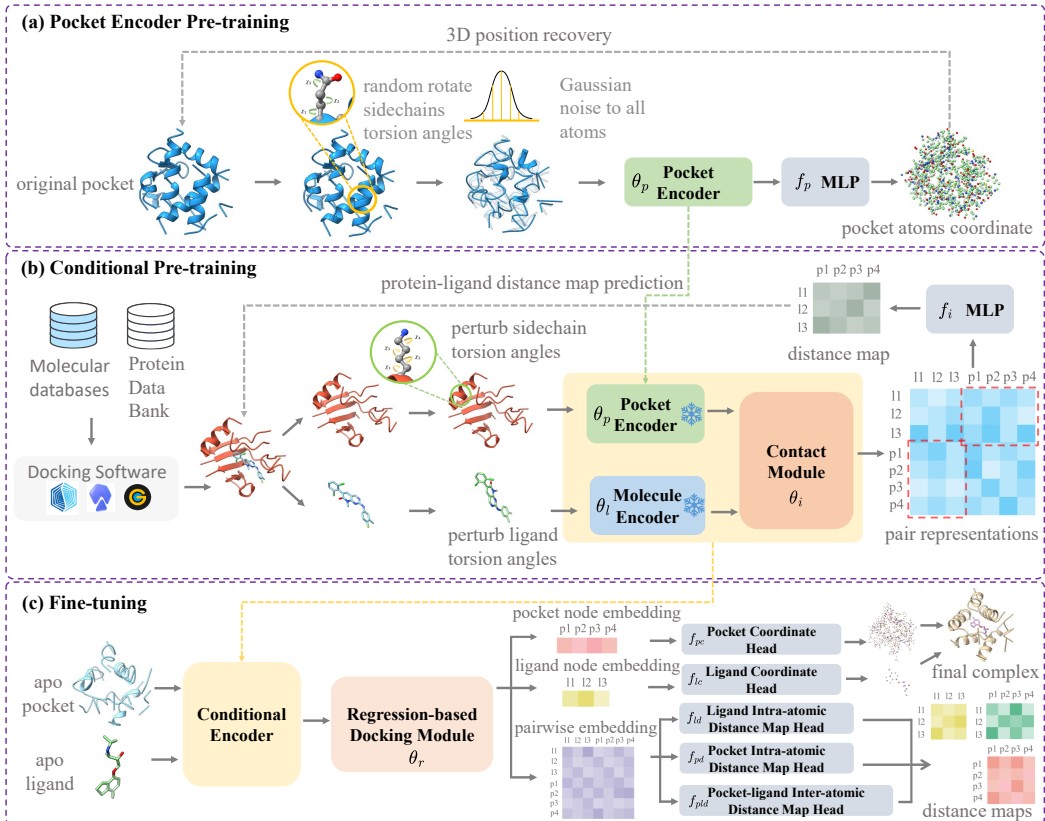

Figure 2: Illustrations of FIGRDock consist of three components. **a**: Apply a combination of noise to the pocket, including perturbed dihedral angles and coordinates, then denoise it to train a pocket encoder that is aware of side chains. **b**: Use coarse structures generated by docking software to learn a conditional pair representation. **c**: Fine-tune on accurate crystal complex structures, using the coarse conditional pair representation to guide the regression-based docking module during the fine-tuning process.

## 3 FIGRDock

In this section, we present our proposed method, FIGRDock. We begin by introducing the notations and formalizing the flexible docking task. We then describe the two key components of FIGRDock: (1) the conditional pair representation learning module, which captures interaction-aware features between the protein and ligand, and (2) the regression-based docking module, which directly predicts the bound complex structure in a single forward pass.

### 3.1 Preliminaries and Problem Formalization

A protein-ligand complex can be represented as $\mathcal{G} = (\mathcal{V}, \mathcal{X})$, where $\mathcal{V}$ represents the set of atom types $v_i$ for the vertices $i$, and $\mathcal{X}$ represents the set of coordinates $x_i$ for each vertex. The complex can be divided into two parts: a ligand and a protein. The ligand part is represented as $\mathcal{G}^l = (\mathcal{V}^l, \mathcal{X}^l)$, and the protein part is represented as $\mathcal{G}^p = (\mathcal{V}^p, \mathcal{X}^p)$. The atom types for the small molecule (ligand) are consistent with those defined by the periodic table (e.g., C, N, O...). However, for the pocket atom types, to model information about side-chain variations, we treat the same element at different positions on the side chain and backbone as different types. For instance, for the element Carbon (C), we distinguish between C (backbone carbonyl), CA (alpha carbon), CB (beta carbon), etc. (Refer to the Appendix 6.1 for the complete list of side-chain atom types).

In the flexible docking setting, the goal is to predict the structural changes of both the protein and the small molecule that occur during the binding process. Given the unbound apo structures of the ligand and the protein, represented as $\mathcal{G}^{l^*} = (\mathcal{V}^l, \mathcal{X}^{l^*})$ and $\mathcal{G}^{p^*} = (\mathcal{V}^p, \mathcal{X}^{p^*})$, respectively, the task is to

predict their bound conformation: $\mathcal{G}^l = (\mathcal{V}^l, \mathcal{X}^l)$ for the ligand and $\mathcal{G}^p = (\mathcal{V}^p, \mathcal{X}^p)$ for the protein. This requires modeling the mutual conformational adjustments that occur upon binding, making the task significantly more complex than rigid docking.

## 3.2 Conditional Pair Representation Learning

Figure 2a illustrates the process for learning conditional pair representations. Initially, the protein pocket and the ligand are processed by two separate pre-trained encoders to obtain initial node representations: $h_p = \theta_p(\mathcal{G}^{p^*})$ and $h_l = \theta_l(\mathcal{G}^{l^*})$. Here, $\theta_p$ and $\theta_l$ denote the encoders for the pocket and the molecule, respectively. For the ligand encoder $\theta_l$, we adopt the pre-trained molecular encoder from Uni-Mol [24].

To pre-train the pocket encoder $\theta_p$, we design a side-chain denoising task using pocket data provided by ProFSA [6]. Specifically, we apply a combined noise scheme to the original pocket $\mathcal{G}^p$: first, we perturb its rotatable dihedral angles, and second, we add Gaussian noise to the coordinates of all its atoms. This process yields the noised pocket $\mathcal{G}^{\tilde{p}}$. The learning objective for the pocket encoder pre-training can be represented as:

$$\mathcal{L}_d = \mathbb{E}_{\mathcal{G}^p, \mathcal{G}^{\tilde{p}}} ||f_p(\theta^p(\mathcal{G}^{\tilde{p}})) - (\mathcal{X}^{\tilde{p}} - \mathcal{X}^p)||_2^2, \tag{1}$$

Among them, $\mathcal{X}^{\tilde{p}}$ and $\mathcal{X}^p$ represent the coordinates of $\mathcal{G}^{\tilde{p}}$ and $\mathcal{G}^p$ respectively, and $f_p$ represents an MLP (Multi-Layer Perceptron) structure, which is used to predict the coordinate noise of the pocket from the pocket representation.

Subsequently, using the initial node representations $h_l$ and $h_p$ obtained from the separate encoders, we learn the conditional pair representation utilizing complex data from SIU [7] (generated by docking software calculations). Specifically, for this stage, we perturb the rotatable dihedral angles within the ligand and the side chains of the pocket in the complex data. This generates noised, approximately apo-like conformations denoted as $\mathcal{G}^{\hat{l}}$ (ligand) and $\mathcal{G}^{\hat{p}}$ (pocket). Let $\mathcal{D}_{pl}$ represent the distance matrix of the holo pocket structure and the small molecule structure within the complex. As shown in Figure 2b, the purpose of the interaction network module $\theta_i$ is to take the noisy conformations $\mathcal{G}^{\hat{p}}$ and $\mathcal{G}^{\hat{l}}$ as inputs, and learn the conditional pair representation by predicting $\mathcal{D}_{pl}$. Specifically, the loss function can be defined as:

$$\mathcal{L}_c = \mathbb{E}_{\mathcal{G}^{\hat{l}}, \mathcal{G}^{\hat{p}}} ||f_i(h_{\hat{pl}}) - \mathcal{D}_{pl}||_2^2, \tag{2}$$

Where $f_i$ represents the MLP utilized for predicting the distance matrix, and $h_{\hat{pl}} = \theta_i(\theta_l(\mathcal{G}^{\hat{l}}), \theta_p(\mathcal{G}^{\hat{p}}))$ represents the conditional pair representation learned by network $\theta_i$. During the training of $\theta_i$, the parameters of $\theta_p$ and $\theta_l$ are kept frozen.

## 3.3 Regression-based Docking Module

As shown in Figure 2c, the regression-based docking module $\theta_r$ takes the unbound (apo) structures $\mathcal{G}^{l^*}$ and $\mathcal{G}^{p^*}$, along with the learned pair representation $h_{pl}$, as inputs to predict the bound (holo) complex structures $\mathcal{G}^l$ and $\mathcal{G}^p$.

To enable direct coordinate regression while capturing both intra- and inter-molecular structural constraints, we construct three distance matrices:

- $\mathcal{D}_l$: the intra-ligand atomic distance matrix,
- $\mathcal{D}_p$: the intra-protein atomic distance matrix,
- $\mathcal{D}_{pl}$: the inter-molecular atomic distance matrix between ligand and protein atoms.

These matrices are computed from the predicted coordinates and serve as targets in our training objective. Specifically, the coordinate prediction loss for the ligand is defined by comparing the predicted intra-ligand atomic distances with the ground truth, encouraging the model to preserve realistic molecular geometry.

$$\mathcal{L}_{ligand} = \mathbb{E}_{\mathcal{G}^{p^*}, \mathcal{G}^{l^*}} (||f_{lc}(\hat{h}_l) - (\mathcal{X}^l - \mathcal{X}^{l^*})||_2^2 + ||f_{ld}(\hat{h}_{pl}) - \mathcal{D}_l||_2^2). \tag{3}$$

Similarly, the coordinate prediction loss for the pocket can be expressed as follows to enforce physically realistic geometry within the binding pocket:

$$\mathcal{L}_{pocket} = \mathbb{E}_{\mathcal{G}^{p^*}, \mathcal{G}^{l^*}} (||f_{pc}(\hat{h}_p) - (\mathcal{X}^p - \mathcal{X}^{p^*})||_2^2 + ||f_{pd}(\hat{h}_{pl}) - \mathcal{D}_p||_2^2). \tag{4}$$

Lastly, the following loss is defined to penalize deviations between the predicted and ground-truth inter-molecular atomic distance matrix across the protein-ligand interface:

$$\mathcal{L}_{interface} = \mathbb{E}_{\mathcal{G}^{p^*}, \mathcal{G}^{l^*}} (||f_{pld}(\hat{h}_{pl}) - \mathcal{D}_{pl}||_2^2). \tag{5}$$

In the above losses, $\hat{h}_l, \hat{h}_p, \hat{h}_{pl} = \theta_r(\mathcal{G}^{l^*}, \mathcal{G}^{p^*}, h_{pl})$ denotes the resulting node embedding of ligand, node embedding of pocket and pairwise embedding encoded by $\theta_i$, respectively. The $f_{lc}, f_{ld}, f_{pc}, f_{pd}$, and $f_{pld}$ represent the head network for the prediction of the coordinate matrix and the distance.

The total regression docking loss is the sum of these three components:

$$\mathcal{L}_r = \mathcal{L}_{ligand} + \mathcal{L}_{pocket} + \mathcal{L}_{interface}. \tag{6}$$

## 4 Experiments

### 4.1 Main Experiments

**Experimental Setup** For pocket encoder pre-training, we use pocket data provided by ProFSA [6] to perform sidechain-aware pre-training. Since the noise-adding process involves perturbing sidechain torsions, we filtered the dataset to remove samples with incomplete sidechains, reducing the total number of samples from 5 million to 4.8 million. The pre-training was conducted on 4 GPUs for 10 epochs with a batch size of 64, taking approximately 6 days to complete.

For conditional pre-training, we pre-train on the SIU [7] dataset, which consists of 5.34 million complex conformations generated by docking software. The training was conducted using 4 A100 GPUs with a batch size of 16, and the pre-training took approximately 20 days to complete.

In the fine-tuning stage, we fine-tune FIGRDock on the commonly adopted PDBbind v2020 dataset[22], which contains 19K crystal complex structures. We employ the time-split of PDB-bind with 17k complexes from 2018 or earlier for training and validation, and 363 test structures from 2019, ensuring consistency with previous works[19, 4]. The input apo ligand conformation is generated using RDKit with a random seed, while the input apo protein structure is predicted by ESMFold [10]. The fine-tuning is performed on 4 A100 GPUs for 100 epochs with a batch size of 16, taking approximately 3 days to complete. Detailed hyperparameters can be found in the Appendix 6.1.

**Evaluation Metric** We evaluate FIGRDock on the PDBbind test set and the PoseBusters V2 [3] test set. The PoseBusters V2 Benchmark is a curated collection of 308 high-quality, drug-like protein–ligand crystal complexes released after 2021, specifically designed to assess docking methods not only in terms of RMSD but also based on chemical and geometric plausibility through RDKit-based quality checks. The primary evaluation metric is the RMSD of Cartesian coordinates. We report the percentage of samples with RMSD below different thresholds, specifically < 2Å and < 5Å for ligands, along with the median RMSD value across all samples. We also report the average runtime to evaluate the model's efficiency. Finally, for the PoseBusters benchmark, we report the PBValid score, which reflects the model's ability to generate chemically and structurally reasonable conformations.

**Baselines** For the PDBbind benchmark, we compare FIGRDock with search-based models SMINA [9] and GNINA [14], which are traditional methods employing scoring functions and search algorithms to effectively explore ligand poses at a considerable computational cost. We also compare FIGRDock with generation model-based pocket-level docking methods, DiffDock-Pocket [17], ReDock [8], and FlexDock [4]. For the PoseBusters V2 benchmark, we measure FIGRDock against search-based models GOLD [21] and VINA [20], generation model-based FlexDock [4], and co-folding models UMol [2] and AlphaFold3 [1].

#### 4.1.1 PDBbind

As shown in Table 1, we compare FIGRDock's performance and runtime with search-based models SMINA [9] and GNINA [14], sidechain flexible models DiffDock-Pocket [17] and ReDock [8],

Table 1: RMSD performance and runtime comparison of different methods on the PDBbind dataset. The best results are highlighted in **bold**. FIGRDock demonstrates a significant advantage in both accuracy and efficiency.

| Models | Holo Crystal Proteins | | | Apo ESMFold Proteins | | | Average Runtime (s) |
|---|---|---|---|---|---|---|---|
| | %<2 ↑ | %<5 ↑ | Med.↓ | %<2 ↑ | %<5 ↑ | Med.↓ | |
| SMINA(rigid) | 32.5 | 54.7 | 4.5 | 6.6 | 22.5 | 7.7 | 258 |
| SMINA | 19.8 | 47.9 | 5.4 | 3.6 | 20.5 | 7.3 | 1914 |
| GNINA(rigid) | 42.7 | 67.0 | 2.5 | 9.7 | 33.6 | 7.5 | 260 |
| GNINA | 27.8 | 54.4 | 4.6 | 6.6 | 28.0 | 7.2 | 1575 |
| DiffDock-Pocket(40) | 49.8 | 79.8 | 2.0 | 41.7 | 74.9 | 2.6 | 61 |
| ReDock(40) | 53.9 | 80.3 | 1.8 | 42.9 | 76.4 | 2.4 | 58 |
| FlexDock | - | - | - | 39.7 | - | 2.5 | 11 |
| FIGRDock | **57.2** | **82.3** | **1.6** | **46.6** | **76.8** | **2.3** | **0.4** |

and all-atom flexible model FlexDock [4]. We report results for both rigid and flexible versions of SMINA and GNINA. Overall, FIGRDock outperforms existing methods in both accuracy and inference efficiency. In terms of RMSD performance, metric %RMSD<2Å is a crucial metric as the predicted structure is considered successful when it meets this criterion. FlexDock [4] is the only generative model that has modeled all atoms, making it the most equitable model for comparison. FIGRDock significantly outperformed FlexDock [4] in metric %RMSD<2Å by nearly 7% (46.6% vs. 39.7%) with apo input. Furthermore, FIGRDock improves upon the previous best-performing method, ReDock [8] in metric %RMSD<2Å, by nearly 4% with both holo and apo input (57.2% vs. 53.9% and 44.6% vs. 42.9%). At the same time, in terms of model efficiency, FIGRDock significantly accelerates inference compared to ReDock [8], achieving over a 100-fold speedup (0.4s vs. 58s). This demonstrates that under the guidance of conditional pair embeddings, the regression-based module, which avoids repetitive sampling and iterative reasoning, not only substantially enhances efficiency but also maintains state-of-the-art accuracy.

### 4.1.2 PoseBusters

On PoseBusters, as shown in Figure 3, rigid docking methods including DeepDock [15], Uni-Mol [24], GOLD [21] and VINA [20] receive holo pockets as input. While flexible docking methods, including FlexDock and FIGRDock use apo input generated by ESMFold. We also report the result that FIGRDock uses holo as input. Finally, co-folding methods like Umol [2] and AlphaFold3 [1] take sequences as input. FIGRDock performs significantly better than FlexDock [4] as well as other deep learning based rigid docking models like DeepDock [15] and Uni-Mol [24]. Compared to search-based methods like GOLD [21] and VINA [20], although FIGRDock with apo input falls slightly behind, it is much faster and takes on a prominently harder task. FIGRDock achieves better performance than GOLD and Vina with holo input. AlphaFold3 [1] significantly outperformed all methods, as it is trained using a larger volume of data. FIGRDock can generate more physically plausible conformations, achieving 99.5% and 96.7% PBValid for apo and holo input. Details of validity checks for the PoseBusters V2 benchmark are deferred to Appendix 6.4.

## 4.2 Ablation study

### 4.2.1 Comparison with ESM embedding

In this study, we compare our approach with the traditional method that uses protein language model–generated embeddings as conditions, as reported in Table 2. 'Without condition' refers to the model trained without any conditional pre-training, while 'With ESM' denotes the use of amino acid–level node embeddings extracted from ESM2 [10]. Our method, FIGRDock, employs the proposed conditional pair representation. All settings use the same network architecture and fine-tuning strategy to ensure a fair comparison.

Results shown in Table 2 demonstrate that our interaction-guided approach provides substantial benefits for molecular docking. First, the significant performance improvement of our method over the

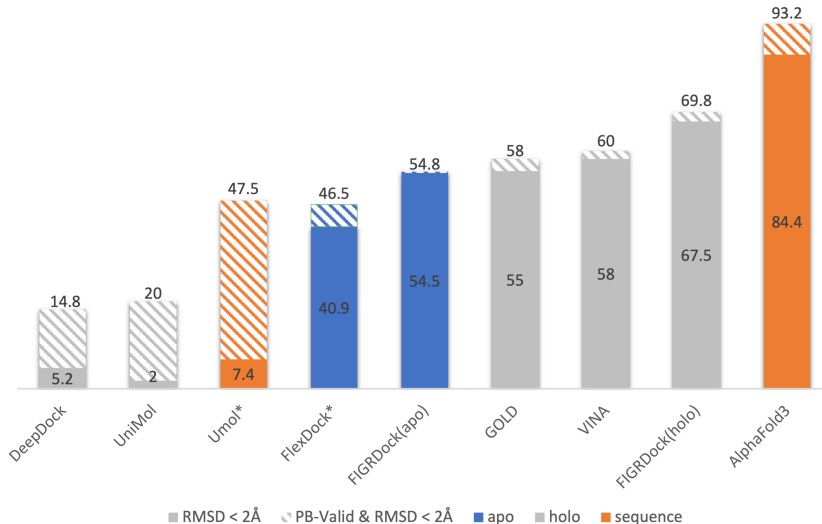

Figure 3: Results of the PoseBusters V2 benchmark with known pockets. FIGRDock outperforms the flexible docking method FlexDock with apo input. Meanwhile, FIGRDock outperforms search-based methods (Gold and Vina) with holo input. For methods marked with *, we demonstrate results reported by the FlexDock paper [4].

Table 2: RMSD performance comparison of different protein representations for docking. 'Without condition' uses no conditional pre-training; 'With ESM' uses ESM2-based residue level node embeddings; FIGRDock employs interaction-aware conditional representations. The best results are in **bold**, showing the advantage of our approach over general protein features.

| Models | Holo Crystal Proteins | | Apo ESMFold Proteins | |
|---|---|---|---|---|
| | %<2 ↑ | Med.↓ | %<2 ↑ | Med.↓ |
| Without condition | 46.2 | 2.2 | 37.8 | 2.8 |
| With ESM | 49.0 | 2.0 | 37.8 | 2.7 |
| FIGRDock | **57.2** | **1.6** | **46.6** | **2.3** |

unguided baseline validates the effectiveness of our conditional learning strategy. Moreover, FIGR-Dock consistently outperforms the 'With ESM' setting across both holo and apo inputs—especially for apo ESMFold proteins, where the success rate (%RMSD<2Å) increases by nearly 9%. This performance gain is particularly notable considering that our approach incurs significantly lower training costs than ESM2. These findings suggest that representations capturing interaction-specific knowledge offer more relevant and efficient guidance for docking tasks compared to general-purpose protein representations.

#### 4.2.2 Impact of Apo Structure Prediction Methods on Docking Performance

Table 3: RMSD performance of FIGRDock models trained on apo structures predicted by different folding methods (AlphaFold2 vs. ESMFold), evaluated on both AlphaFold2- and ESMFold-predicted apo test sets.

| Models | Apo AlphaFold2 Proteins | | Apo ESMFold Proteins | |
|---|---|---|---|---|
| | %<2 ↑ | Med.↓ | %<2 ↑ | Med.↓ |
| FIGRDock(Training by AlphaFold2) | 47.5 | **2.1** | 36.7 | 2.6 |
| FIGRDock(Training by ESMFold) | **48.1** | **2.1** | **46.6** | **2.3** |

Apo protein conformations can be predicted using either ESMFold or AlphaFold2, both of which generate 3D protein structures from amino acid sequences. However, prediction accuracy varies

between methods, and few studies have explored how different predicted apo structures influence downstream docking performance. Here, we evaluate the robustness of our model when provided with apo structures predicted by different folding algorithms. This experiment is critical to determine whether our model's performance depends on specific conformational inputs.

To this end, we constructed two dataset variants for training and evaluation. In addition to the main experimental setup using ESMFold, we created a variant based on AlphaFold2-predicted apo structures. The data processing and split strategy follows the same protocol as FABFlex [23]. We trained two models separately using apo structures predicted by ESMFold and AlphaFold2, and evaluated each model on both ESMFold- and AlphaFold2-predicted test sets. Results are shown in Table 3.

We observe that when the training and testing apo structures come from the same folding method, FIGRDock performs well in both cases. Interestingly, the model trained on ESMFold data generalizes well to AlphaFold2-predicted test structures. In contrast, the model trained on AlphaFold2 data performs poorly on the ESMFold-predicted test set. We hypothesize that this is due to AlphaFold2's higher prediction accuracy, which may result in less noisy apo conformations in the training set, thereby limiting the model's ability to generalize to noisier samples in the ESMFold test set.

### 4.2.3 Scaling Study of Pre-training Data

In this section, we investigate how the scale of pre-training data influences model performance by varying the dataset size used for conditional pre-training, ranging from 0 (as noted earlier, this corresponds to the 'Without condition' setting in Table 2, i.e., no conditional pre-training) to 5 million samples (the full SIU [7] dataset). Figure 4 demonstrates a positive correlation between the scale of pre-training data and docking performance across all evaluation metrics, under both apo and holo test scenarios. This consistent improvement with increasing data scale validates the effectiveness and flexibility of our framework, as well as its potential to benefit from even larger datasets. Due to computational constraints, we currently report full experiments only on the 5-million-sample setting. Future work can explore larger-scale datasets to further enhance performance.

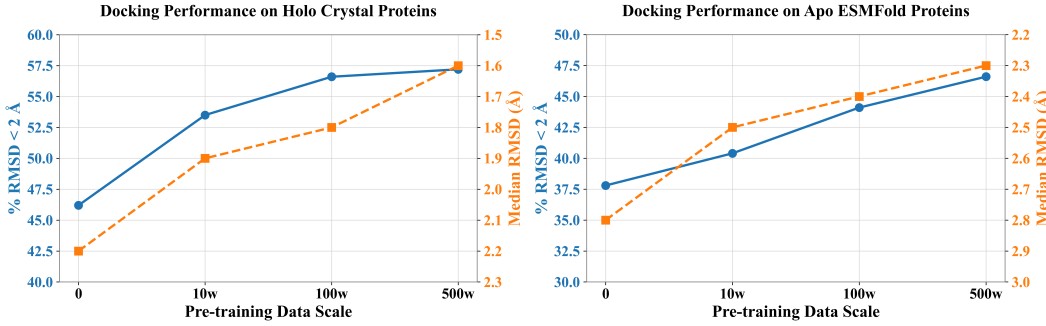

Figure 4: Comparison of docking performance with different scales of pre-training data. Larger pre-training datasets lead to better performance across both holo (left) and apo (right) settings.

## 5   Conclusion

In this work, we present FIGRDock, a fast and accurate regression-based framework for flexible molecular docking. Unlike mainstream generative methods that rely on repetitive sampling, scoring, and pre-trained protein embeddings, FIGRDock adopts an interaction-aware conditional representation to guide direct regression of protein-ligand complex structures. By decoupling the learning of interaction patterns from the final docking prediction, FIGRDock achieves high docking accuracy with a single forward pass, significantly improving inference efficiency. Extensive experiments on both holo and apo settings demonstrate that FIGRDock not only outperforms previous diffusion-based models in accuracy but also achieves nearly 100× faster inference. These results highlight the promise of regression-based docking under interaction-guided supervision and open new directions for efficient and scalable structure-based drug design.

## Acknowledgments and Disclosure of Funding

This work is supported by Beijing Academy of Artificial Intelligence and Beijing Frontier Research Center for Biological Structure Fundings.

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

# 6 Technical Appendices and Supplementary Material

## 6.1 Implementation Details

The training of FIGRDock consists of three stages: Pocket encoder pre-training, conditional pre-training, and fine-tuning. The hyperparameter settings used in all stages are listed in Table 4.

Table 4: Hyperparameter settings for Pocket Pretraining, Conditional Pretraining, and Fine-tuning stages.

|  | Pocket Pretraining | Conditional Pretraining | Fine-tuning |
|---|---|---|---|
| Batch Size | 64 | 16 | 16 |
| Training Epochs | 10 | 12 | 100 |
| Learning Rate | $1 \times 10^{-4}$ | $3 \times 10^{-4}$ | $3 \times 10^{-4}$ |
| LR Scheduler | `polynomial_decay` | `polynomial_decay` | `polynomial_decay` |
| Warmup Ratio | 0.01 | 0.06 | 0.06 |
| Optimizer | Adam | Adam | Adam |
| Weight Decay | $1 \times 10^{-4}$ | 0 | 0 |
| GPU Number | 4 | 4 | 4 |

For pocket encoding, to enhance the model's sensitivity to side-chain variations during the docking process, we treat atoms with the same elemental type but different structural roles, such as backbone versus side-chain atoms, as distinct atom types. In particular, for side-chain atoms, we define a comprehensive set of atom types to capture their structural specificity, including:

```
C, CA, CB, CD, CD1, CD2, CE, CE1, CE2, CE3, CG, CG1, CG2, CH2, CZ, CZ2, CZ3,
N, ND1, ND2, NE, NE1, NE2, NH1, NH2, NZ, O, OD1, OD2, OE1, OE2, OG, OG1, OH,
                                  SD, SE
```

We employ a Transformer-based architecture to encode molecular and protein structures. Specifically, the encoding schemes for atom types and 3D positions, along with the design of the Transformer layers, are adopted from Uni-Mol[24].

## 6.2 Evaluating Model Generalization Beyond Structural Memorization

The docking structures used in pre-training provide only coarse-grained structural information. Initially, we did not consider their similarity to the test set in our experiments. To further investigate whether the model demonstrates true generalization ability rather than memorizing recurring patterns between the training and test data, we analyzed the structural similarity between the SIU pre-training set and the PDBbind test set.

To minimize potential data leakage, we removed all training samples whose structural similarity to any test sample exceeded 0.5. After this filtering, the training set retained 5,018,392 entries. We then repeated the pre-training and fine-tuning procedures using this filtered dataset.

Table 5: RMSD comparison of FIGRDock models trained with and without structural-similarity filtering.

| Models | Holo Crystal Proteins | | Apo ESMFold Proteins | |
|---|---|---|---|---|
| | %<2 ↑ | Med.↓ | %<2 ↑ | Med.↓ |
| FIGRDock | 57.2 | **1.6** | **46.6** | 2.3 |
| FIGRDock (similarity-filtered) | 57.2 | 1.7 | 45.7 | **2.2** |

As shown in Table 5, after removing highly similar structures from the training data, FIGRDock maintains nearly identical performance compared to the original model. Although the proportion of predictions with RMSD < 2 Å slightly declines on the apo test set, the model still achieves competitive results and substantially outperforms all baselines. This result indicates that the model has indeed learned transferable and generalizable representations, rather than simply memorizing structural patterns seen during training.

## 6.3 Visualized Examples

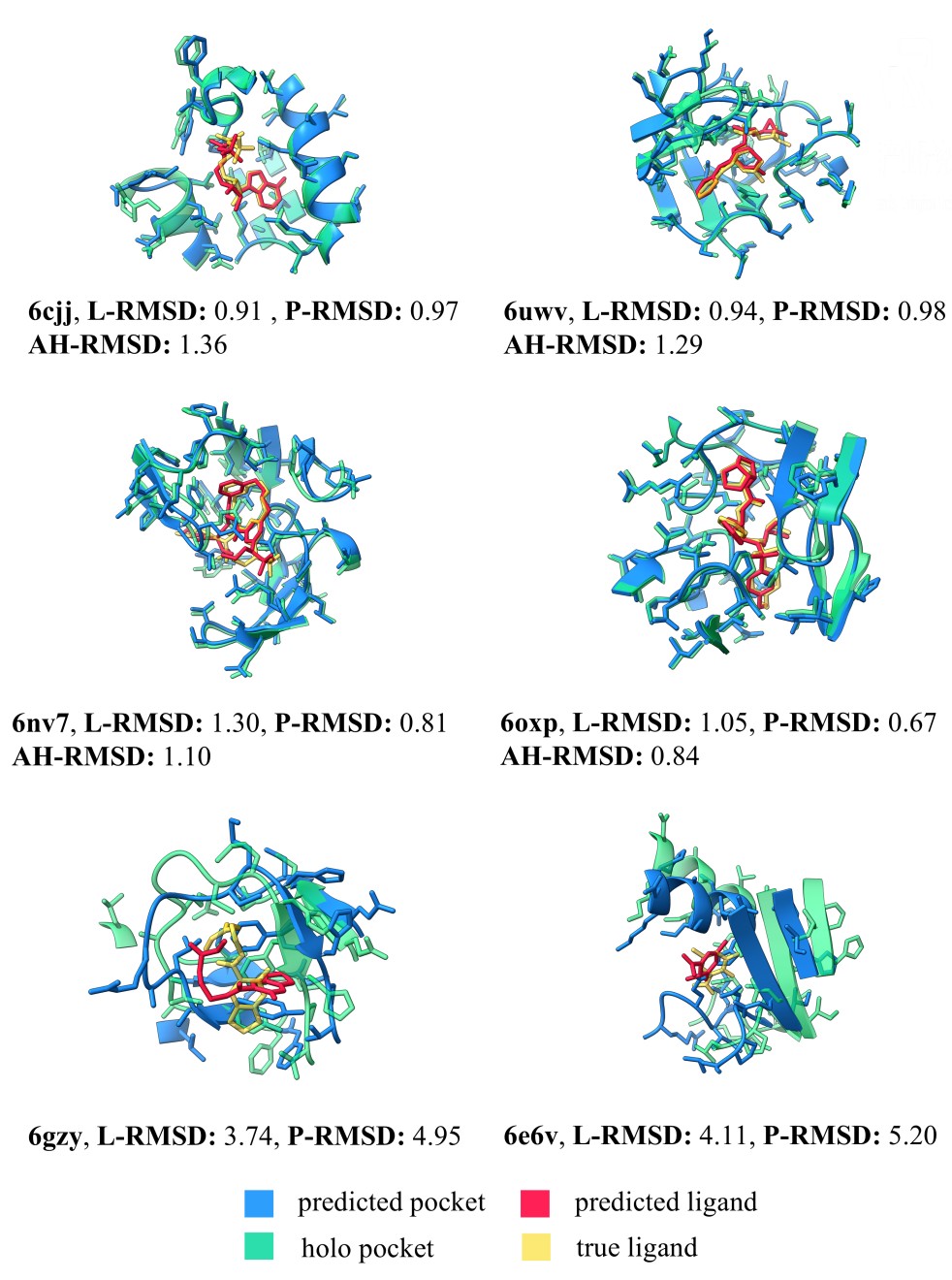

**6cjj**, **L-RMSD:** 0.91 , **P-RMSD:** 0.97
**AH-RMSD:** 1.36

**6uwv**, **L-RMSD:** 0.94, **P-RMSD:** 0.98
**AH-RMSD:** 1.29

**6nv7**, **L-RMSD:** 1.30, **P-RMSD:** 0.81
**AH-RMSD:** 1.10

**6oxp**, **L-RMSD:** 1.05, **P-RMSD:** 0.67
**AH-RMSD:** 0.84

**6gzy**, **L-RMSD:** 3.74, **P-RMSD:** 4.95

**6e6v**, **L-RMSD:** 4.11, **P-RMSD:** 5.20

predicted pocket    predicted ligand

holo pocket    true ligand

Figure 5: Visualized examples of complexes 6cjj, 6uwv, 6nv7, 6oxp, 6gzy and 6e6v in PDBbind test dataset. L-RMSD measures the RMSD between the predicted ligand and the ground-truth ligand. P-RMSD denotes the RMSD between the predicted pocket and the ground-truth holo pocket. AH-RMSD represents the RMSD between the input apo pocket and the ground-truth holo pocket.

## 6.4 Additional Experiment Results

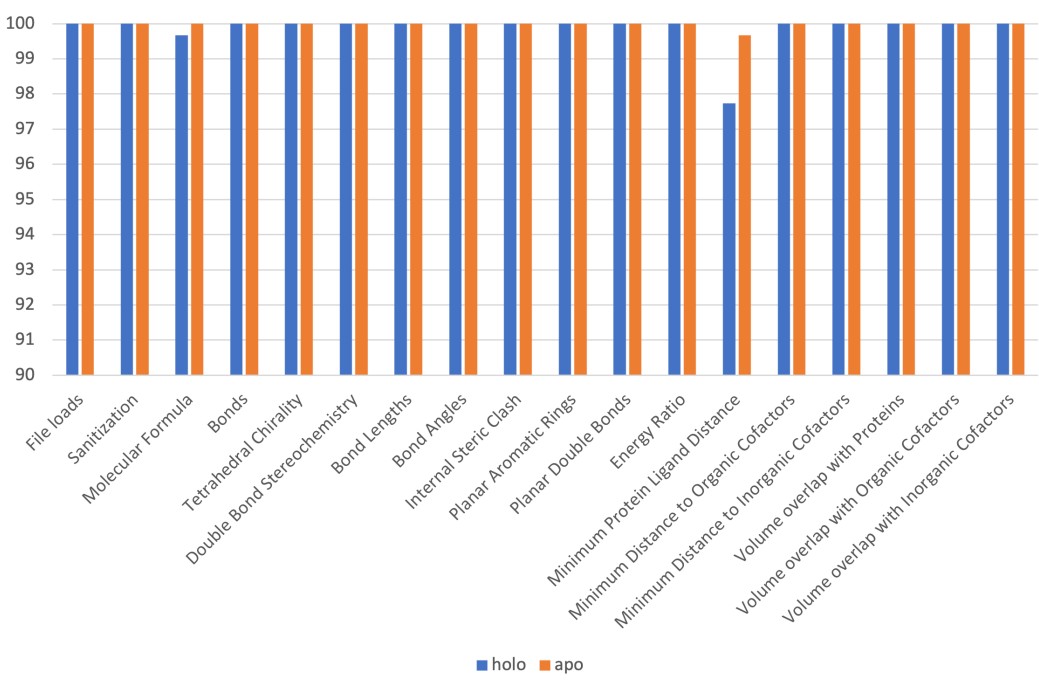

Figure 6: Detailed plausibility checks for predictions by FIGRDock on PoseBusters V2 benchmark with holo and apo input. FIGRDock achieves 99.5% and 96.7% PBValid for apo and holo input, generating physically reasonable conformations.

