# OpenReview forum: "FIGRDock: Fast Interaction-Guided Regression for Flexible Docking"
_NeurIPS.cc/2025/Conference — NeurIPS 2025 poster_

### Official Review · Reviewer_q2Sc · 2025-06-30

**Clarity:** 3
**Significance:** 2
**Originality:** 3
**Rating:** 4
**Confidence:** 4

**Summary:**

This paper introduces FIGRDock, a regression framework for one-shot flexible protein-ligand docking. The core idea is a two-stage training strategy. First, the model is pre-trained on large datasets  to learn conditional pair representations. Second, this pre-trained model is fine-tuned on a smaller set of high-quality experimental crystal structures (PDBbind).  Tested on the PDBbind and PoseBusters, FIGRDock outperforms baselines such as FlexDock and ReDock.

**Questions:**

**Suggestions/Questions:**
1. Providing a similarity analysis between pre-training dataset and testing set.
2. Providing a comparison between more powerful baselines, such as DiffBind-FR and HelixDock.
3. Besides the task, what are the main difference between FIGRDock and previous pre-training docking methods, such as CarsiDock?

I'm willing to raise my score if these **suggestions/questions** are solved.

**Ethical Concerns:**

["NO or VERY MINOR ethics concerns only"]

**Final Justification:**

My concerns have been solved.

**Limitations:**

Yes

**Quality:**

3

**Strengths And Weaknesses:**

**Strengths**：
1. The paper is generally well-written and easy to follow.
2.  FIGRDock outputs baselines (such as FlexDock, ReDock, and Smina) on the PDBbind and PoseBusters test set


**Weaknesses**:
1. Lack of Technical Originality. The strategy of pre-training on large datasets of computationally docked structures has been explored in prior work (HelixDock, CarsiDock). The pre-training strategy of ligand encoder and pocket encoder are also previous Uni-Mol and ProFSA.
2. The paper omits several highly relevant and competitive baselines in the flexible docking space. For example, DiffBind-FR.
3. There is a major concern regarding the unfair comparison of the reported results. The model is pre-trained on the SIU dataset, which is derived from structures in the PDB. The PDBbind test set is a time-split subset of the PDB. It is highly probable that the molecular systems (or highly similar ones) in the PDBbind test set are present in the massive SIU pre-training dataset. While the authors use a time-split for fine-tuning, they do not mention any filtering of the pre-training data based on the test set. This data contamination would mean the model has essentially "seen" the test examples during pre-training, making the strong performance unsurprising and casting doubt on the model's true generalization ability to novel targets.

---

> ### Author Rebuttal · Authors · 2025-07-30
>
> > Providing a similarity analysis between pre-training dataset and testing set
>
> Thank you for your valuable suggestions. In our initial experiments, we did not filter out overlapping PDB IDs between the pre-training dataset and the PDBbind test set, as the structures in the pre-training data were generated by docking software and are generally considered noisy and approximate.
>
> To further investigate this concern, we analyzed the overlap between the pre-training datasets (at different scales: 100K, 1M, and 5M/full dataset) and the PDBbind test set in terms of both exact PDB ID matches and sequence-level homology. Specifically, we calculated the number of shared PDB IDs as well as the proportion of proteins with sequence similarity greater than 90% and 50%. The results are summarized in the following table:
>
> | Pre-training Scale | PDB ID Overlap | Seq. Sim. >90% | Seq. Sim. >50% |
> |--------------------|----------------|----------------|----------------|
> | 100K               | 0              | 0              | 0              |
> | 1M                 | 0.60%          | 4.10%          | 5.70%          |
> | 5M (full)          | 0.80%          | 5.30%          | 8.60%          |
>
> We observe that in the 100K setting, there is no PDB ID overlap and essentially no sequence similarity to the test set. Yet, even under this setting, FIGRDock yields noticeable performance improvements (e.g., RMSD < 2: 37.8% → 40.4% on the apo set), indicating that the gains are not purely due to memorization.
>
> To further examine whether the improvements stem from memorization or generalization, we conducted an ablation study under the 1M pre-training setting due to time constraints during the rebuttal period. We performed two filtering strategies: (1) removing all exact PDB ID overlaps, and (2) further removing all proteins with sequence similarity greater than 50%. The results are as follows:
>
> | PDBBind (APO)                    | RMSDs < 2Å ↑ | Median RMSD ↓ |
> |----------------------------------|--------------|----------------|
> | Without condition                | 37.8         | 2.8            |
> | 100k                              | 40.4         | 2.5            |
> | 1M                             | 44.1         | 2.4            |
> | 1M (erase overlap)             | 43.5         | 2.5            |
> | 1M (erase > 50% seq sim)       | 42.6         | 2.5            |
>
> - We found that removing overlapping PDB IDs and similar proteins (seq. sim > 50%) led to a drop in performance.
> - However, even after removing all proteins with >50% sequence similarity, the model still outperformed both the no-pretraining baseline and the 100K pre-training setting.
>
> These results indicate that while similar proteins in the pre-training set do contribute to performance improvements, they are not the sole reason for the gains. Our model has indeed learned generalizable representations rather than simply memorizing training data.
>
> In the future, we plan to conduct the same filtering experiments on the full pre-training dataset and update the corresponding results in the paper. We sincerely appreciate your thoughtful suggestion.
>
>
> > Providing a comparison between more powerful baselines
>
> We present additional results comparing our method with DiffBind-FR and HelixDock on the PDBbind and PoseBusters V1 benchmarks, as shown in the table below. Our approach demonstrates a clear performance advantage over DiffBind-FR:
>
> | Method           | Task | PDBbind (RMSD < 2Å) | PoseBusters V1 (RMSD < 2Å) | PoseBusters V1 (RMSD < 2Å & PB-valid) |
> |------------------|----------------------|----------------------|---------------------------|-------------------------------------|
> | DiffBind-FR      | flexible docking | 51.2%                | 48.1%                     | 44.4%                               |
> | FIGRDock (ours)  | flexible docking | 57.2%                | 63.3%                     | 60.7%                               |
> | HelixDock        | rigid docking | 90.1%                | 85.6%                     | 85.2%                               |
>
>
> Since both DiffDock-FR and our method fall under the category of flexible docking, they are not directly comparable to HelixDock, which is based on rigid docking.
>
> > Besides the task, what are the main difference between FIGRDock and previous pre-training docking methods, such as CarsiDock?
>
> Although both our method and CarsiDock incorporate a pre-training stage, our approach targets a different and more challenging task scenario—flexible docking—in which both the protein and the small-molecule ligand undergo conformational changes. And CarsiDock is conducted on rigid docking, only considering molecule flexibility. Our pre-training strategy is carefully designed to align with our final application. Specifically, we introduce a pocket encoder that captures side-chain information to better model protein flexibility. Furthermore, during the noise-adding process, we apply perturbations to protein side chains to simulate the conformational transition from the apo to holo state.
>
> > The difference with Uni-Mol and ProFSA
>
>
> Our pre-training strategy differs from UniMol primarily in the design of the noise mechanism. Specifically, we apply a hybrid noise to the pocket, combining side-chain perturbations with Gaussian noise, whereas UniMol uses noise sampled from a uniform distribution. We consider the side-chain perturbation noise to be closer to physical protein dynamics. Additionally, we extend the atom type vocabulary by treating atoms of the same chemical type but located at different positions within protein side chains as distinct types. Although we utilize the ProFSA dataset, our training objectives and strategy are  different from the contrastive learning approach used in ProFSA.

---

> ### Comment · Area_Chair_Y7Bi · 2025-08-05
>
> Reviewer q2Sc,
>
> Can you please check the rebuttal and add your comments?
> Note that reviewers must participate in discussions with authors before submitting “Mandatory Acknowledgement”.
>
> AC

---

> ### Comment · Reviewer_q2Sc · 2025-08-05
>
> Dear AC,
> Thank you for the reminder. I had clicked "Mandatory Acknowledgement" under the mistaken assumption that it was only to confirm I had read the rebuttal. I'll keep in discussion with the authors and my response is below.

---

> > ### Comment · Reviewer_q2Sc · 2025-08-05
> >
> > Thank you for your response. Most of my concerns have been addressed.
> >
> > However, my concern regarding the techinical novelty still remain, for the following reasons:
> >
> > i) The main contribution of this work appears to be its pre-training strategy. However, the effectiveness of pre-training on docking data has already been established in prior docking methods (HelixDock, CarsiDock).
> >
> > ii) All pre-training strategies already existing. For the ligand encoder, the method directly adopts the existing Uni-Mol. For the pocket encoder, the side-chain denoising task used for pre-training is a straightforward idea to conceive and implement, as current diffusion-based flexible docking methods like DiffBind-FR denoise side-chain rotational angles when traning. What's more, the strategy of combining side-chain perturbations with Gaussian noise is also previous work[1].
> >
> > iii) The authors claim the new atom type vocabulary as a contribution. However, the paper lacks an ablation study to support this claim. Consequently, it is unclear whether this component has any significant impact on the final performance.
> >
> > **Suggestion**:
> > 1. Providing a further clarfication about this work's contribution.
> >
> > 2. An ablation study about the new atom type vocabulary is useful for demonstrating its effectiveness.
> >
> > [1] Pre-training with fractional denoising to enhance molecular property prediction.

---

> ### Author Response · Authors · 2025-08-06
>
> Dear reviewer q2Sc:
>
> Thank you very much for your prompt response and constructive feedback. We sincerely hope that our clarifications have adequately addressed your concerns. We remain at your disposal for any further information or explanation that may assist in a more comprehensive evaluation.
>
> > Providing a further clarification about this work's contribution.
>
> Thank you for your suggestions. We would like to provide further clarification regarding the contributions of our work:
>
> 1. Difference from HelixDock and CarsiDock
>
> While both HelixDock and CarsiDock have demonstrated the effectiveness of pre-training in __rigid docking__ tasks, our setting is fundamentally different — we focus on __flexible docking__. For this purpose, we designed a pre-training strategy specifically tailored to model the conformational changes of protein side chains, which is the core principle behind our pre-training approach.
>
> 2. Regarding existing pre-training strategies
>
> First, we would like to clarify that the novelty of a paper should not be solely judged by whether it uses an existing or easy-to-implement pre-training strategy. Many influential works adopt well-known and relatively straightforward strategies — such as next-token prediction [1], contrastive learning [2], and masked language modeling [3] — yet their novelty and impact are still widely recognized.
>
> Second, although our strategy could broadly be categorized as a denoising strategy, there are key differences between our approach and existing works. For example, while DiffBind-FR also applies noise to side chains, it (like DiffDock and DiffDock-Pocket) performs denoising on a special manifold — a product space — whereas we recover the original distance matrix directly. Moreover, our pre-training is not a diffusion process — the noise magnitude is independent of any timestep.
>
> Compared to Frad, which designs a hybrid noise for molecular generation (comprising both torsional and Gaussian coordinate noise), our approach is distinct in both purpose and implementation. Although FigrDock uses similar noise types to train a pocket encoder, Frad’s goal is to ensure equivalence to learning a force field through a fractional denoising strategy that only denoise the Gaussian noise component. In contrast, our goal is to simulate protein side chain flexibility, and our denoising target includes both torsional and Gaussian noise components (as illustrated in Figure 2a of our paper).
>
> We also notice that Reviewer qhP6 shares our perspective and acknowledges the novelty of our work, stating: “_The technical contributions/choices adopted by the paper are sound, especially in the context of flexible docking_,” and “_Pre-training a pocket encoder for docking with a denoising task is novel, even though similar strategies have been used for small molecule representation learning_.”
>
>
> In summary, our main novelty lies in addressing the inefficiency of generative methods in flexible docking. We propose a new framework that learns interaction-aware conditions, which are then used to guide the fine-tuning of a regression-based model. This approach compensates for the limitations in regression accuracy, enabling fast and accurate prediction of docking poses.
>
>
> [1]: MolGPT: Molecular Generation Using a Transformer-Decoder Model
>
> [2]: 3D Infomax improves GNNs for Molecular Property Prediction
>
> [3]:Evolutionary-scale prediction of atomic-level protein structure with a language model.
>
>
> > An ablation study about the new atom type vocabulary is useful for demonstrating its effectiveness.
>
> Thank you for your suggestion. To address this concern, we conducted a comparative experiment. Specifically, we introduced a new setting using a basic atom vocabulary, where atoms of the same type at different side-chain positions are not distinguished (for example, both CA and CB are treated as generic carbon atoms, "C"). This setting is referred to as the `basic vocab`, while the one used in our paper with an extended vocabulary that distinguishes side-chain positions is referred to as the `sidechain vocab`.
> We re-ran the conditional pre-training and fine-tuning pipeline under the basic vocab setting using the 1M SIU dataset. (_Please note that this new experiment could not be fully completed within the remaining rebuttal period. Fortunately, we had planned such an ablation study earlier—though the results were not included in the final paper—and thus we are now able to use it to answer your question._)The results are summarized in the table below.
>
> | Vocabulary Setting | Holo: RMSD < 2 (%) | Apo: RMSD < 2 (%) |
> |--------------------|--------------------|--------------------|
> | `Basic Vocab`        | 54.3               | 42.9               |
> | `Sidechain Vocab`    | __56.6__               | __44.1__               |
>
> As shown in the table, the sidechain vocab outperforms the basic vocab, demonstrating the effectiveness of our proposed atom-type vocabulary design.

---

> > ### Comment · Reviewer_q2Sc · 2025-08-06
> >
> > Thank your for the rebuttal, i'll raise my score to 4.

---

> > > ### Author Response · Authors · 2025-08-09
> > >
> > > Dear reviewer q2Sc,
> > >
> > > We greatly appreciate your careful review of our rebuttal. It is encouraging to know that our responses addressed your concerns. Thank you for your constructive engagement, and we wish you all the best!

---

### Official Review · Reviewer_qhP6 · 2025-07-01

**Clarity:** 3
**Significance:** 2
**Originality:** 3
**Rating:** 4
**Confidence:** 4

**Summary:**

This paper introduces a regression-based approach, FIGRDock, for pocket-based flexible docking. Key to the success of regression-based approaches is a precise understanding of protein-ligand interactions and associated distances, which are then used by the structure module to then predict the final structure. The precise understanding of protein-ligand interactions is achieved through a novel pre-training stage, utilizing pocket-reconstruction and distance-map prediction tasks. Empirical results showcase the improved performance of FIGRDock over previous baselines.

**Questions:**

Questions are marked with a **Questions** identifier in the Weaknesses section.

**Ethical Concerns:**

["NO or VERY MINOR ethics concerns only"]

**Final Justification:**

The authors addressed most of my concerns during the rebuttal and I'm happy to raise my score.

**Limitations:**

Limitations are not discussed, and should be included. For example, generative models align well with the experimental uncertainty associated in the docking task, and have an implicit ability to correct their predictions. This is however missing in regression based models.

**Quality:**

2

**Strengths And Weaknesses:**

### **Strengths**:

* Accurately modelling protein flexibility in docking is an important yet understudied problem. The paper takes a step in the right direction in that regard, with a very computationally efficient method.

* The submission is clearly written, well-organized, and the technical details and contributions of the paper are well-explained to the reader.

* The technical contributions / choices adopted by the paper are sound, especially in the context of flexible docking. To my best knowledge, pre-training a pocket-encoder for docking with a denoising task is novel, even though similar strategies have been used for small molecule representation learning. Empirical investigations confirm the effectiveness of these choices (some caveats remain, see Weaknesses).


### **Weaknesses / Clarifications:**

**Technical Contributions:**

1. The main technical contribution of the paper lies in the conditional pre-training stage, where both the pocket encoder, and pair-representation between the ligand and protein is learnt.

**Question**: For the pocket encoder denoising task, do you only use a single noise scale, or is it a combination of noise scales as in diffusion models?

2. For the distance-map prediction task, it appears as though a regression paradigm is used by directly predicting distances. A commonly used strategy across co-folding approaches is the distogram prediction which insteads utilizes a classification perspective.

**Questions:**

* Did the authors experiment with both regression and classification settings, and if yes, have any insights on when one paradigm would be preferred over the other?
* How do the authors ensure training is stable across different distance scales in the regression setting?

**Empirical Results:**

1. From the ablation results, it is quite clear that the greatest improvement in performance comes from the conditional pre-training strategy (~9% improvement). This improvement emerges from one of the following reasons:

* The pre-training data from ProSFA and SIU datasets contain a lot of diversity in pocket structures and binding modes, and allow the model to learn flexible interaction patterns, improving generalization.

* The interaction patterns in the pre-training data largely mimic patterns seen in the test set (this is a known issue with the time-split of PDBBind), and the increased performance is from additional memorization.

In my experience, such large improvements are often from sort of memorization rather than true generalization. Given the mounting evidence of similar issues also for co-folding models such AlphaFold3 and Boltz-1 [1], my suspicion for the improved performance is from the latter reason.

This is a critical issue as the main utility of docking models is to provide fast and accurate predictions across the proteome. If the conditional pre-training stage only offers a better route to memorization, the technical contributions of the paper are immediately thin.

**Question:** Is it possible to construct an overlap score similar to [1], between the 5 million poses in SIU dataset and the PDBBind split test set? The distogram of the similarity to the closest training set point should give a clear indication of whether memorization was happening.

2. For flexible docking, one is also often interested in predicting accurate protein structures, given the downstream utility in binding affinity prediction.

**Question:** Can the authors also compare the pocket-aligned All-Atom RMSD < 1 of their method against other baselines (wherever available)? This should help understand if the pocket-encoder pre-training endows the model with some understanding of the protein conformational space.

3. The learnt interactions are only evaluated from a geometric perspective.

**Question:** Can the authors also evaluate the protein-ligand interaction recovery metrics from [2] to see if key physico-chemical patterns are preserved?

4. This is minor, but most docking papers claim that fast approaches are needed to support the demands of virtual screening. However, there are a few papers that showcase the effectiveness of active-learning style approaches in finding reasonable hits for virtual screening, which makes computational efficiency (in comparison to accuracy) a less of a bottleneck [3, 4]

I have currently recommended the paper as Borderline Reject. If there is very little evidence for memorization from the pre-training stage, I am happy to raise my score. On the other hand, if the evidence is overwhelmingly in favor of memorization, I will keep my rating.

**References:**

[1]: Have protein-ligand co-folding methods moved beyond memorization?\
[2]: Assessing Interaction Recovery of Predicted Protein-Ligand Poses\
[3]: Thompson Sampling─An Efficient Method for Searching Ultralarge Synthesis on Demand Databases\
[4]: New Trends in Virtual Screening

---

> ### Author Rebuttal · Authors · 2025-07-31
>
> > For the pocket encoder denoising task, do you only use a single noise scale, or is it a combination of noise scales as in diffusion models?
>
> Sorry about the confusion. We only use a single noise scale in the pocket encoder denoising task. This approach is much simpler than using a combination of noise scales as in diffusion models. Neither extra modules for time embedding nor complicated loss function are needed. Exploring the use of multiple noise scales could be an interesting direction for future work.
>
> >  Did the authors experiment with both regression and classification settings
>
> We did not explore the classification setting and instead adopted a regression setting.  The main reason we directly adopt the regression setting is that we believe it provides more fine-grained supervision signals compared to classification.
>
>
> > How do the authors ensure training is stable across different distance scales in the regression setting
>
> We indeed employ a strategy for stabilizing the training process. A predicted distance is ignored for gradient calculation when it goes beyond a threshold of 8 Å. Following the practice of Uni-Mol, we apply a mask to zero out the loss for atom pairs whose ground-truth distance is greater than 8Å. These pairs do not contribute to the gradient during training.
>
> >  Response to memorization or true generalization
>
> Thanks for your suggestions. The docking structures used in pre-training only provide coarse-grained structural information. Initially, we did not consider their similarity to the test set in our experiments. To further investigate whether the model demonstrates memorization or generalization, we analyzed the similarity between the SIU test set and the PDBbind test set.
>
> Specifically, we examined subsets of the pre-training data used in Section 4.2.3 (i.e., the 100K and 1M subsets and the 5M full set) and measured their overlap with the test set in terms of PDB ID, protein sequence homology (with similarity metrics such as sequence identity), and calculated the proportion of SIU samples that fall into these overlapping categories. The analysis results are presented in the table below.
>
>
> | Pre-training Scale | PDB ID Overlap | Seq. Sim. >90% | Seq. Sim. >50% |
> |--------------------|----------------|----------------|----------------|
> | 100K               | 0              | 0              | 0              |
> | 1M                 | 0.60%          | 4.10%          | 5.70%          |
> | 5M (full)          | 0.80%          | 5.30%          | 8.60%          |
>
> We found that under the 100K pre-training setting, there was no significant overlap or high similarity between the pre-training and test sets. Yet, the model still demonstrated generalization capability — for example, the percentage of structures with RMSD < 2 on the apo set increased from 37.8% to 40.4%.
>
> However, for the 1M and 5M pre-training settings, we did observe some degree of overlap and similarity with the test set. To further examine whether the improvements were due to memorization or generalization, we conducted an ablation study under the 1M pre-training setting (Due to time constraints during the rebuttal period, we were only able to afford the experimental cost of this subset setting). We applied two filtering strategies:
> - Removing all exact PDB ID overlaps.
> - Further removing all proteins with sequence similarity > 50%.
>
> After obtaining the filtered dataset, we re-conducted the pretraining and fine-tuning processes, yielding the following results:
>
>
> | Pre-training settings\ PDBBind (APO)                    | RMSDs < 2Å ↑ | Median RMSD ↓ |
> |----------------------------------|--------------|----------------|
> | Without condition                | 37.8         | 2.8            |
> | 100k                              | 40.4         | 2.5            |
> | 1M                             | 44.1         | 2.4            |
> | 1M (erase overlap)             | 43.5         | 2.5            |
> | 1M (erase > 50% seq sim)       | 42.6         | 2.5            |
>
> From the above table, we can conclude that:
> - Removing overlapping PDB IDs and similar proteins (sequence similarity > 50%) did lead to a drop in performance.
> - However, even after removing all proteins with >50% sequence similarity, the model still outperformed both the no-pretraining baseline and the 100K pre-training setting.
>
> These findings suggest that while similar proteins in the pre-training set do contribute to performance gains, they are not the sole reason for the improvements. Our model has indeed learned generalizable representations, rather than simply memorizing the training data.
>
> As for the similarity score between the pre-training data and test data mentioned in paper [1], we attempted to compute it using the runs-n-poses benchmark approach for our own pre-training and test datasets. However, we found that a necessary step in the data preparation process—run_structure_qc—would require approximately one week to complete, which exceeds the time constraints of the rebuttal period. Therefore, we conducted the alternative experiment described above to provide supporting evidence for our claim.
>
>
> In the future, we plan to perform the same filtering experiments on the full pre-training dataset and update the corresponding results in the paper. We sincerely appreciate your thoughtful and constructive feedback.
>
> [1]:  Have protein-ligand co-folding methods moved beyond memorization?
>
>
> >  Can the authors also compare the pocket-aligned All-Atom RMSD < 1 of their method against other baselines
>
>
> We evaluated FIGRDock's performance on PDBBind benchmark with pocket-aligned All-Atom RMSD < 1 and we compare with baselines mentioned in our paper. And the results are as follows:
>
> | | Pocket-aligned AA RMSD %<1↑   |
> |-------------------------|---------------------|
> |DiffDock-Pocket | 32.4 |
> |FlexDock | 39.8 |
> |FlexDock | 41.7 |
> |FIGRDock(ours)| 41.5 |
>
> FIGRDock achieved comparable result with baseline models in terms of pocket RMSD metric. This result indicates that the pocket-encoder pre-training facilitates the model's acquisition of a certain level of comprehension regarding the conformational space of proteins.
>
>
> > Can the authors also evaluate the protein-ligand interaction recovery metrics from [2] to see if key physico-chemical patterns are preserved?
>
> Thank you for the suggestion. We carefully studied the methodology described in the paper and applied it to evaluate the protein-ligand interaction recovery rates of our PoseBusters test results. Our evaluation results are shown in the figure below. The results for all methods except FIGRDock are taken directly from Figure 2 of the original paper.
>
> | Category                 | Method                | RMSD < 2 | RMSD < 2 && PBValid | RMSD < 2 && PBValid && PLIF-valid @ 50% | RMSD < 2 && PBValid && PLIF-valid @ 100% |
> |--------------------------|-----------------------|----------|----------------------|------------------------------------------|-------------------------------------------|
> | Classical Docking        | GOLD                  | 75.7     | 70.9                 | 63.3                                     | 28.4                                      |
> | Classical Docking        | HYBRID                | 74.3     | 45.6                 | 37.0                                     | 16.8                                      |
> | Classical Docking        | FRED                  | 53.1     | 34.6                 | 30.1                                     | 14.9                                      |
> | Machine Learning Method  | FIGRDock              | 70.5     | 68.3                 | 42.7                                     | 12.8                                      |
> | Machine Learning Method  | DIFFDock-L            | 47.6     | 24.1                 | 21.4                                     | 7.0                                       |
> | Machine Learning Method  | RosettaFold-AllAtom   | 19.1     | 7.3                  | 2.9                                      | 1.5                                       |
> | Machine Learning Method  | Umol                  | 24.5     | 2.9                  | 0.7                                      | 0.3                                       |
>
>
> As we can see, FIGRDock outperforms other machine learning methods across all metrics, including RMSD, PBValid, and interaction recovery rate. However, there is still a performance gap between FIGRDock and the classical docking method GOLD in terms of RMSD and PLIF-related metrics. This finding is consistent with the conclusions reported in the original study.
>
>
>
> > About limitation
>
> We agreed with you that generative models can account for uncertainty in the docking task and it has the ability to correct its prediction through iterative sampling. We will further discuss these limitations in our final version.

---

> > ### Comment · Reviewer_qhP6 · 2025-08-04
> > **Thank you for your Response**
> >
> > Dear authors,
> >
> > Thank you for your response. Most of my questions have been addressed.
> >
> > My concerns still remain mostly in the context of memorization vs generalization. As pointed out in my questions, the similarity score between SIU and PDBBind should have been constructed from a structural similarity metric and not a sequence-based one. One should ideally consider something like the TM-Score of the pocket and the volume overlap / other metrics (as suggested in [1]) and look at the joint similarity.
> >
> > While I can see that performance is improved even on a filtered pre-training set, it is still unclear to me that this translates to a structural similarity metric.
> >
> > Thus, I will continue to maintain my rating.

---

> > > ### Author Response · Authors · 2025-08-08
> > >
> > > Dear reviewer qhP6:
> > >
> > > Thank you for your constructive feedback. We have provided further detailed responses regarding the structural similarity to better demonstrate the model's generalization ability in response to your comments. As the rebuttal period is approaching its end, we would like to confirm whether our responses have sufficiently addressed your concerns. We look forward to your further feedback.

---

> > > > ### Comment · Reviewer_qhP6 · 2025-08-08
> > > > **Thank you for your Response**
> > > >
> > > > Dear authors,
> > > >
> > > > Thank you for your response.
> > > >
> > > > The analysis above is still missing the ligand, since the similarity score in the Runs N Poses paper is a combination of both the protein and the ligand.
> > > >
> > > > That said, I appreciate the analysis you have already run, and it is good to see the trend not overwhelmingly in favor of memorization.
> > > >
> > > > I'm happy to raise my score to 4, but request the complete analysis to be included should the paper be accepted.

---

> > > > > ### Author Response · Authors · 2025-08-09
> > > > >
> > > > > Dear reviewer qhP6,
> > > > >
> > > > > We are glad to know that our earlier responses have addressed most of your concerns. Your acknowledgment and insightful feedback—particularly regarding the similarity score experiment and the generalization vs. memorization analysis—are invaluable. We will carry out the suggested experiment and include the additional analysis in the camera-ready version, if it is accepted. Thank you again for your time, expertise, and thoughtful suggestions.

---

> ### Author Response · Authors · 2025-08-05
>
> Dear reviewer qhP6:
>
> Thank you for your suggestion. To address your comment, we have extended our analysis by incorporating TM-score–based structural similarity metrics for between the SIU dataset and the PDBbind test set. Due to time constraints, the TM-score calculations were performed on a 100k subset of the SIU dataset and the entire PDBbind test set, which contains 355 samples.
>
> The TM-scores were computed using the `tmtools` [1] package. For each test sample in the PDBbind set, we selected the maximum TM-score obtained against all 100k SIU samples. Owing to rebuttal policy constraints, we are unable to provide a figure analogous to the one in [2]. Instead, we report the proportion of samples with RMSD < 2 across different TM-score intervals in the table below:
>
>
> | TM-score Interval | [0, 0.2) | [0.2, 0.4) | [0.4, 0.6) | [0.6, 0.8) | [0.8, 1.0] |
> |-------------------|----------|-------------|-------------|-------------|------------|
> | Sample Count      | 0        | 279         | 72          | 4           | 0          |
> | RMSD < 2 Ratio    | 0        | 0.5341      | 0.5139      | 1.0000      | 0          |
>
>
> As noted in [3], a TM-score greater than 0.5 typically indicates significant structural similarity. As shown in the accompanying table, the majority of test samples (279 out of 355) fall within the (0.2, 0.4) TM-score range, with an average TM-score of 0.35—suggesting limited overall similarity. Nevertheless, our pretraining on the 100k SIU subset still delivers a clear performance gain over the baseline without pretraining (e.g., RMSD < 2: 46.2% vs. 53.5%; the original results are presented in Figure 4 of the paper).
>
> We further analyzed the distribution of RMSD < 2 across different TM-score intervals. Interestingly, the proportion of successful predictions does not exhibit a monotonically increasing trend with structural similarity, contrary to what might be expected from Figure 1 in [2]. For example, although the [0.6, 0.8] interval achieves a perfect RMSD < 2 rate of 1.0—likely indicative of memorization—this group contains only 4 samples. In contrast, the [0.2, 0.4] interval includes the majority of samples and achieves a higher prediction accuracy than the [0.4, 0.6] interval (0.5341 vs. 0.5139), contributing most to the overall average performance (0.535). These results suggest that our FigRDock method demonstrates strong generalization capabilities, rather than merely relying on memorization.
>
> [1]: tmtools: https://github.com/jvkersch/tmtools/tree/main
>
> [2]: Have protein-ligand co-folding methods moved beyond memorization?
>
> [3]: TM-Score: https://zhanggroup.org/TM-score/

---

### Official Review · Reviewer_4soF · 2025-07-02

**Clarity:** 2
**Significance:** 2
**Originality:** 2
**Rating:** 3
**Confidence:** 4

**Summary:**

This paper proposes a regression-based method for flexible docking, pretrained on computational data and fine-tuned on crystal data. The method achieves comparable results in terms of both accuracy and efficiency.

**Questions:**

1.	Section 3 would be strengthened by a more detailed explanation of the modules. Could the authors elaborate on their implementation or architecture?
2.	For a fair and direct comparison, the baseline models should be trained and evaluated on the same datasets as FIGRDock. The current comparison is weakened by this discrepancy.
3.	Figure 1 appears to be redundant, as it repeats information already present in Figure 2. Additionally, the framework diagram should be annotated with the mathematical notations for each module (e.g., labeling the "Ligand Coordinate Head" with f_lc) to improve clarity and connect the diagram to the text.

**Ethical Concerns:**

["NO or VERY MINOR ethics concerns only"]

**Final Justification:**

Although the author has provided a detailed response, the core innovation of the paper still needs to be improved.

**Limitations:**

Yes

**Quality:**

2

**Strengths And Weaknesses:**

Strengths:
1.  This work applies a regression-based method, typically used for rigid docking, to the task of flexible docking and achieves results comparable to those of diffusion-based models, offering a new alternative for future research.
2.  This work provides the insight that a "one-shot approach demands precise interaction modeling" and proposes learning this interaction through a pre-training paradigm on data generated by docking software.
3.  This work significantly improves the efficiency of flexible docking prediction compared to diffusion-based models, making it more practical for real-world applications.
Weaknesses:
1.  The novelty of this work is limited, as the primary innovation is the "Conditional Pre-training" process, with no modifications to the prediction framework itself.
2.  Section 3, which details the FIGRDock method, is poorly written. For example, it lacks formal mathematical equations and structural details for key modules such as the "Contact Module" and "Regression-based Docking Module," providing only the loss functions.
3.  The performance of FIGRDock is highly reliant on the SIU pre-training set, while the baseline models were trained only on PDBbind. This discrepancy in training data leads to an unfair comparison.

---

> ### Author Rebuttal · Authors · 2025-07-30
>
> > Response to the novelty
>
> Our work focuses on the complex and challenging task, flexible molecular docking, providing a fast and accurate regression-based framework. FIGRDock distinguishes itself from previous methods with the following contributions:
> - New Perspective: precise interaction modeling matters. Mainstream generative methods that rely on repetitive sampling, scoring, and pre-trained protein embeddings, lack efficiency for real-world application. Prior regression-based approaches offer a more efficient way for molecular docking but often struggle to make accurate prediction with one-shot inference. FIGRDock takes into account both efficiency and accuracy by decoupling the learning of interaction patterns from the final docking prediction.
> - New Methodology: a simple yet effective pre-training and fine-tuning framework. We utilized SIU dataset, which contains enormous computational complex data, overcoming the limitation of the small number of crystal data. Further, our novel pre-training strategy, including pocket reconstruction and distance map prediction tasks, is effective for learning the interaction patterns. To the best of our knowledge, FIGRDock is the first regression-based approach solving all-atom level flexible docking problem in a pre-training and fine-tuning fashion.
> - New Performance. Experiments show FIGRDock outperforms all the baselines on the standard benchmark PDBbind (Section 4.1.1). Notably, it even achieves a nearly 9% relative improvement over the previous SOTA model ReDock. On PoseBusters benchmark (Section 4.1.2), FIGRDock performs significantly better than FlexDock as well as other deep-learning-based rigid docking models, and achieves better performance than search-based methods GOLD and Vina with holo input. Moreover, ablation study (Section 4.2.3) highlights the scalability potential of our method.
>
>
> And we notice that Reviewer qhP6 agrees that our work is novel, as it mentions, "the precise understanding of protein-ligand interactions is achieved through a novel pre-training stage" and "pre-training a pocket-encoder for docking with a denoising task is novel".
>
> > More details of the Method
>
> Thank you for the suggestions. We illustrate more details about the network architecture here and will revisit the methodology described in Section 3 of the paper in the future.
> We adopt a Transformer-based architecture in our framework, with the basic layer structure following Uni-Mol due to its simplicity. The input to each layer consists of node-level embeddings $ h_s $ and pair-wise embeddings $ q $.
>
> Each Transformer layer performs the following operations:
>
>  (1) compute the query, key, and value vectors $Q$, $K$, $V$ for each head based on the atomic representations $h_i^l$;
>
>  (2) compute attention weights with the pairwise bias $ q_{ij}^l $;
>
> (3) update the atomic representations using:
> $h_i^{l+1} = \text{FFN}(\text{MultiHeadAttention}(h_i^l, q_{ij}^{l}))$
> and
>  (4) update the pairwise representations using:
> $ q_{ij}^{l+1} = q_{ij}^l +   \{ \frac{Q_i^{l,h} \cdot (K_j^{l,h})^\top}{\sqrt{d}} \mid h \in [1, H] \} $
>
> where $l$ denotes the layer index and $h$ denotes the head index among $H $ total attention heads.
> For both the pocket and ligand encoders, the node-level embeddings are initialized based on atom types, while the pairwise-level embeddings are initialized using the Euclidean distances between atom pairs.
>
> For the input to the contact module, the node-level embedding is the concatenated output of the ligand and pocket encoders.
>
> As for the pairwise-level embeddings, denoted as $q^0_{\text{PL}}$, we initialize them with zeros and then populate them with the values from $q^0_{\text{PP}}$ and $q^0_{\text{LL}}$ at the corresponding positions as follows:
>
> $
> q^0_{\text{PL}}[:L_p, :L_p] = q^0_{\text{PP}}, \quad
> q^0_{\text{PL}}[-L_m:, -L_m:] = q^0_{\text{LL}}
> $
>
> Here, $L_p$ and $L_m$ denote the number of atoms in the pocket and the ligand, respectively. The updated node- and pairwise-level embeddings produced by the contact module are then used as input to the docking regression module.
>
> >  Response to redundancy of Figure 1
>
> Thanks for your feedback! Instead of redundancy, we believe that Figure 1 and Figure 2 play different roles in presenting our method. Figure 1 is designed to enable readers to have a quick understanding of the flexible docking problem tackled by FIGRDock. As shown in Figure 1, FIGRDock mainly comprises two modules: a conditional encoder and a regression-based docking module. The conditional encoder is pre-trained on computational complex data to provide a interaction guidance. The regression-based docking module is fine-tuned with accurate experimental complex data to perform fast interaction-guided regression for flexible docking. When it comes to Figure 2, we dive into the implementation details of these two modules and present complete workflow of FIGRDock, starting from the pre-training strategy to the final output of different heads. Therefore, we believe that these two figures do not conflict with each other and there is no redundancy.
>
> > Response to framework diagram
>
> Thank you for pointing it out. As suggested, we will add the mathematical notations for each module and ensure their consistency with the text.
>
> > Response to the pre-training of other baselines
>
> We would like to clarify that the goal of the method proposed in our paper is to improve both the efficiency and accuracy of flexible docking. The pre-training stage learns interaction representations that, when combined with a regression-based approach, enable fast and accurate structure prediction, outperforming previous generative baselines. While it is possible to adapt generative approaches by incorporating the pre-trained representations as conditional guidance to further improve generation accuracy, this direction falls beyond the scope of the current study.

---

> > ### Comment · Reviewer_4soF · 2025-08-05
> >
> > I have read the author's response, and I'm grateful for the author's clarification. I will adjust my score to 3.

---

> > > ### Author Response · Authors · 2025-08-09
> > >
> > > Dear reviewer 4soF,
> > >
> > > Thank you very much for taking the time to review our rebuttal. We sincerely appreciate your thoughtful acknowledgment of our work and your valuable suggestions. We wish you all the best!

---

### Official Review · Reviewer_bFKc · 2025-07-02

**Clarity:** 3
**Significance:** 3
**Originality:** 3
**Rating:** 5
**Confidence:** 4

**Summary:**

This paper introduces FIGRDock, a new flexible docking framework pretrained on synthetic protein-ligand complexes and fine tuned to complexes derived from laboratory experiments.  FIGRDock takes apo pocket and ligand as inputs and uses regression to match the intra-ligand and intra-protein changes as well as the interfacial interactions between protein and ligand.  The resulting trained model is much faster and more accurate than typical diffusion-based or search-based docking methods, though less accurate than AF3 and its variants.

**Questions:**

Could the authors also run GOLD/VINA starting with apo structures in figure 2?  Also, why use ESMFold and not AF2 for the apo structures---is that simply a result of the analysis in paragraph 4.2.2 or were there additional considerations?

Could the authors test their hypothesis in page 9 paragraph 2 about the origin of the lack of generalization for the model trained on AF2 structures by perhaps augmenting the dataset with artificially noised structures?

What is the x-axis notation in Fig 4?

**Ethical Concerns:**

["NO or VERY MINOR ethics concerns only"]

**Final Justification:**

The authors addressed my main concerns although the origin of the generalization is still not very clear.  I assume that the authors will further improve during the remaining review period and I am willing to optimistically raise my score in response.

**Limitations:**

yes

**Paper Formatting Concerns:**

Elements of Fig2 should have larger fonts.

**Quality:**

3

**Strengths And Weaknesses:**

This paper demonstrates that a simple, MSA-free approach of machine-learned docking can reach high performance, competitive with traditional software used in drug discovery.  This is an important recent area of research and the solution by the authors is notably faster during inference than any of the previous flexible docking approaches.  The presentation of the design of the model and its training protocol are clear.

The main weakness of this work relates to the challenge in evaluating it for its intended purpose: helping drug discovery.  Although the results on time-split datasets are useful and desirable for the sake of comparison to past approaches, it would be useful to also present different, harder data splits and, perhaps most importantly, to stratify the evaluation by similarity to the training set data, for example as done in Fig 1A,B,C of Škrinjar et al, 2025 (https://doi.org/10.1101/2025.02.03.636309).  This is critical, given that AF3 (superior to FIGRDock in accuracy though much slower) struggles with novel molecules and pockets that matter the most in practical contexts.  Addressing this generalization gap would significantly improve this work.

---

> ### Author Rebuttal · Authors · 2025-07-30
>
> > Response to the weakness
>
> Thank you for your suggestions. We conducted an evaluation of the model's generalization ability by examining the similarity between the pre-training data and the test data. Due to the limited time available during the rebuttal period, we removed samples from the 1M pre-training dataset that have a sequence similarity greater than 50% with the test set. The results are shown in the figure below.
>
> | PDBBind (APO)                      | RMSDs < 2Å ↑ | Median RMSD ↓ |
> |-----------------------------------|--------------|----------------|
> | Without condition                 | 37.8         | 2.8            |
> | 100k                               | 40.4         | 2.5            |
> | 1M                                | 44.1         | 2.4            |
> | 1M (erase > 50% seq sim)          | 42.6         | 2.5            |
>
> We observed that although removing similar samples leads to a slight drop in performance, the model trained on the filtered 1M dataset still outperforms both the setting pre-training with 100K dataset and the setting without pre-training. This indicates that the model maintains strong generalization ability. In future work, we will further validate these findings using the full dataset and include additional results in the paper.
>
>
>
> > GOLD/VINA starting with apo structures
>
> We test GOLD/VINA with apo structures as inputs on PoseBusters V2 benchmark. As shown in the table below, our approach significantly surpasses traditional search-based methods, GOLD and VINA.
> | Method   | RMSD < 2Å | RMSD < 2Å & PB-valid |
> |-------------|----------|----------------------|
> | GOLD (apo inputs)    | 10.4%    | 9.4%                 |
> | VINA (apo inputs)    | 13.7%    | —                    |
> | FIGRDock (apo inputs)| 54.8%    | 54.5%                |
>
> Note: The PB-valid metric for VINA requires further validation due to unresolved technical problems; this analysis is ongoing.
>
> > why use ESMFold and not AF2 for the apo structures
>
> We apologize if there was any confusion. The apo structures predicted by ESMFold are intentionally adopted to make a fair comparison with baselines.  The experiment in Section 4.2.2 shows that different cofolding methods used to generate apo data can lead to varying performance on downstream tasks. Previous methods typically test with ESMFold-predicted apo structures. To ensure consistency with previous work, we also employ ESMFold-predicted apo structures for testing.
>
> >  Response to hypothesis in page 9 paragraph 2
>
> The reason we proposed this hypothesis was that the RMSD of the apo protein pockets predicted by AlphaFold2 compared to the holo form is smaller than that predicted by ESMFold. Therefore, the model finetuned with AF2 data took on an easier task and failed to generalize to noisier ESMFold generated structures. To further validate this hypothesis, we attempted to augment the structures predicted by AlphaFold2 by introducing random perturbations to pocket side-chain dihedrals during the fine-tuning process. The experimental results are as follows:
>
> | Method                                      |   Apo AlphaFold2 Proteins                   | Apo ESMFold Proteins           |
> |---------------------------------------------|-------------------------|----------------------|
> |                                             | %<2 ↑ &nbsp;&nbsp;&nbsp;&nbsp; Med.↓    | %<2 ↑ &nbsp;&nbsp;&nbsp;&nbsp; Med.↓    |
> | FIGRDock(Training by AlphaFold2)            | 47.5 &nbsp;&nbsp;&nbsp;&nbsp;&nbsp;&nbsp;&nbsp;&nbsp; 2.1      | 36.7 &nbsp;&nbsp;&nbsp;&nbsp;&nbsp;&nbsp;&nbsp;&nbsp; 2.6      |
> | FIGRDock_sc_aug(Training by AlphaFold2)     | 44.9 &nbsp;&nbsp;&nbsp;&nbsp;&nbsp;&nbsp;&nbsp;&nbsp; 2.2      | 34.4  &nbsp;&nbsp;&nbsp;&nbsp;&nbsp;&nbsp;&nbsp;&nbsp; 2.9      |
>
> Although the experimental results deteriorated after adding noise, we speculate that this is because the way we added noise was too crude and naive, and it could not simulate the complex low-energy conformational changes of proteins in reality. In order to dive into this problem, we plan to add more refined noise, for example sampling side-chain conformations from rotamer libraries. But due to limited time, we cannot present results during the rebuttal period. Thank you for the question you raised. We will continue to explore this issue in the future.
>
> > Response to notation in Figure 4
>
> Sorry about any unclear explanation for Figure 4. The x-axis notation of Figure 4 refers to different data scales we used for pre-training, while the y-axis notation represents docking performance, which includes two types of metrics, the percentage of ligand RMSD below 2 Å and the Median RMSD. Therefore, Figure 4 depicts how good the docking performance is as the data scale increases.
>
> > Response to fonts of elements in Figure 2
>
> Thanks for your valuable feedback! Due to NeurIPS policy "Because of technical complications, we need to stop supporting the global response with PDF", we cannot update our paper PDF immediately. And we will adjust the font sizes in Figure 2 for better readability.

---

> > ### Comment · Reviewer_bFKc · 2025-08-06
> > **Thank you for your comments.**
> >
> > I thank the authors for addressing my main concerns.  Based on the comments by the authors I expect the revised paper to be slightly improved and I have thus updated my score accordingly to 5.

---

> > > ### Author Response · Authors · 2025-08-09
> > >
> > > Dear reviewer bFKc,
> > >
> > > Thank you sincerely for revisiting our rebuttal and taking the time to review it. We are delighted that our clarifications have resolved your concerns, and we truly value your engagement in this process. Wishing you all the best!

---

### Note · Authors · 2025-08-12

Dear Chairs and Reviewers,

Thank you for your dedicated efforts throughout the review process. We sincerely appreciate the thoughtful and constructive feedback from all reviewers. __We are pleased that our rebuttal successfully addressed their concerns, leading all reviewers to raise their scores in recognition of our responses__.


Below we summarize the key improvements and clarifications we made during the rebuttal phase, reflecting our extensive efforts to address reviewers’ concerns and further enhance the quality and clarity of the paper:

__Supplementary Experiments__
  1. Conducted experiments analyzing sequence and structural similarities between the pre-training data and test sets to demonstrate the generalization ability of FIRGDock. (bFKc, qhP6, q2Sc).
  2. Added experiments with sidechain perturbations on AF2-generated apo structures(bFKc).
  3. Added an ablation study on the new atom type vocabulary to demonstrate its effectiveness (q2Sc).

__More Comparisons and Additional Benchmarks__
  1. Added evaluation experiments on protein–ligand interaction recovery metrics (qhP6).
  2. Added a comparison of the pocket all-atom RMSD < 1Å metric with other baselines (qhP6).
  3. Added gold/vina apo test results on PoseBuster v2 (bFKc).

__Clarifications and More Detailed Explanations__
  1. Clarified why we used ESMFold instead of AF2-generated apo structures (bFKc).
  2. Provided further clarification and explanation of Figure 4 (bFKc).
  3. Clarified our contributions and explained differences from previous methods (4soF, q2Sc).
  4. Added more details about the network architecture (4soF).
  5. Clarified the distinct roles of Figures 1 and 2 (4soF).
  6. Clarified that we use a single noise scale instead of a noise schedule for the pocket encoder denoising task (qhP6).
  7. Clarified that we use only the regression setting for the denoising task, and explained the measures taken to ensure stable training in this setting (qhP6).

Regarding the invaluable suggestion from reviewers to revisit the paper, we will include the following in the camera-ready version if it is accepted:
1. Add structural similarity analysis content and a discussion of the limitation that the regression cannot model the experimental uncertainty associated with the docking task (qhP6).
2. Adjust the font size of Figure 2 for better readability (bFKc) and add mathematical notations for each module (4soF).

Thank you for your time and expertise!

Sincerely,

The Authors

---

### Decision · Program_Chairs · 2025-09-17

**Decision:**

Accept (poster)

**Comment:**

Overall, this is a borderline paper. Reviewers raised several concerns regarding the paper, including:
- Technical novelty and contribution
- Evaluation fairness and generalization
- Methodological details

The rebuttal phase was constructive, as the authors provided detailed clarifications and conducted additional experiments, which led to an increase in reviewer scores.

After careful consideration of the reviews and rebuttal, I recommend acceptance. While the novelty is incremental in some respects, the paper makes a meaningful contribution to the flexible docking domain by combining efficiency, accuracy, and scalability.